# The Impact of Schoolyard Greening on Children’s Physical Activity and Socioemotional Health: A Systematic Review of Experimental Studies

**DOI:** 10.3390/ijerph18020535

**Published:** 2021-01-11

**Authors:** Jean C. Bikomeye, Joanna Balza, Kirsten M. Beyer

**Affiliations:** 1PhD Program in Public and Community Health, Institute for Health and Equity, Medical College of Wisconsin, 8701 Watertown Plank Rd., Milwaukee, WI 53226, USA; jbikomeye@mcw.edu (J.C.B.); jbalza@mcw.edu (J.B.); 2Division of Epidemiology, Institute for Health and Equity, Medical College of Wisconsin, 8701 Watertown Plank Rd., Milwaukee, WI 53226, USA

**Keywords:** green space, nature, schoolyard greening, physical activity, socioemotional health, mental health, child health, experimental design, pre-post study design, health equity

## Abstract

Access to green schoolyards (schoolyards designed with greenery and natural elements to create a park-like environment, as opposed to asphalt-based playgrounds) are associated with many benefits for students, including improvements in physical and mental health. While many studies examining these associations are cross-sectional, some feature experimental designs that offer the possibility of causal inference. In this review, we looked at experimental studies that examine the impact of schoolyard greening on measures of physical activity and socioemotional health in children. Four electronic databases (Ovid Medline, PsycINFO, Scopus and Greenfile) were searched, and from 1843 articles retrieved, 6 articles met the inclusion criteria. Examination of the eligible studies revealed a general consensus on the positive impact of schoolyard greening on both physical activity and socioemotional health outcomes for students, suggesting that schoolyard greening is a viable intervention in reducing the health equity gaps and improving children’s health regardless of their racial or ethnic backgrounds or residential neighborhood socioeconomic status. Further experimental research on this topic should elucidate how educators, administrators, policy makers, and other stakeholders can harness the benefits of schoolyard greening to improve the health and well-being of children in their communities.

## 1. Introduction

Research has shown that access to green space has beneficial effects on physical and mental health. Benefits for physical health include: improved self-perceived general health [1], reduced blood pressure [2], reduced cardiovascular-disease-related mortality in adults [3], and improved quality of life [4]. In addition to physical health, access to green space is associated with many positive socioemotional health (SEH) outcomes such as improved mental well-being measured through increase in global life satisfaction [5]; fewer symptoms of depression, anxiety, and stress [6]; reduced stress and recovery from mental fatigue [6,7,8,9]; and increased happiness [5]. Green space also provides neighborhoods with recreational and social engagement opportunities such as walking for leisure [10] and other physical activity (PA) [11,12].

The observed improvement in health outcomes associated with exposure to green spaces is especially prominent in children [13]. In a systematic review of 12 pediatric studies, green space was found to be associated with improved mental well-being, overall health, and cognitive development in children [14]. Other literature shows that green space promotes attention restoration [15], cognitive functioning, behavioral functioning and physiological well-being in children [15,16,17], including reduced likelihood of childhood obesity [18]. Greening has been shown to promote supportive social groups, promote self-discipline, moderate stress, improve behaviors and symptoms of attention deficit hyperactivity disorder in children [14,19,20,21], and improve students’ development of emotional and behavioral regulatory skills [22]. Exposure to green space has even been shown to increase one’s potential to earn more in a lifetime. In a 2019 study, after adjusting for individual and neighborhood confounders and spatial autocorrelation, Browning & Rigolon found that over a 30-year career, children growing up in census tracts with the most vegetative covers still earned on average a cumulative $28,000 more than children growing up in tracts with the least cover [23].

In addition to direct health benefits, greening has been associated with the reduction of traffic-related air pollutants [24], global greenhouse gas emissions [25] improved environmental quality, and better population health [26]. The improvement in environmental quality from green space leads to many health benefits including reductions in respiratory diseases—like asthma due to air pollution—resulting in direct healthcare cost savings of about $13 billion per year [25]. In addition to health care cost savings, greening has been associated with financial savings for schools [25]. Greening is a cost-effective way for schools to enhance student learning, reduce health and operational costs, and, ultimately, increase school quality and competitiveness [25]. Schoolyard greening presents an opportunity to increase children’s exposure to nature [27], while also offering environmental benefits such as stormwater control, air pollution filtration, and reduction of heat islands [25], thus providing a significant local contribution to the global efforts on climate change adaptation and mitigation [28,29,30].

Unfortunately, children are increasingly growing up in areas with limited access to nature [27]. Numerous barriers contribute to this reduction, such as the need to walk long distances to access green space [31] and parental safety concerns [32]. Children from low-income families growing up in those relatively poor urban neighborhoods are especially impacted [27,31,32]. Growing up in urban environments with minimal green space leads to reduced opportunities for engagement in positive behaviors such as PA [27]. Greening schoolyards can help in bridging that health equity gap by providing natural and safe spaces for play opportunities for school children living in urban, low-income neighborhoods, through access to environments that offer opportunities for PA [27,33], reducing stress [6,15,34,35,36], improving attention restoration [35,37], promoting social well-being [38], enabling focus, competence building, and formation of supportive social groups [39]. Every child should have access to equitable play opportunities regardless of where they live or their parental socioeconomic status. Schoolyards can help in reducing those equity gaps because children spend a considerable amount of time every day at schools with recess scheduled in their school day calendars. 

While many studies support the association between schoolyard greening and health benefits in students, few studies offer experimental evidence, and these studies have not been systematically reviewed. Herein we conduct a review of experimental studies of the impact of schoolyard greening on two health outcomes: physical activity (PA) and socioemotional health (SEH). The SEH outcomes consider measures of both children’s social interactions and their mental well-being.

## 2. Methods

This systematic review follows the Preferred Reporting Items for Systematic Reviews and Meta-Analyses (PRISMA) [40] guidelines for systematic reviews. We considered only studies published in the English language.

### 2.1. Literature Search

In the original systematic literature search, four databases were used: Ovid Medline, PsycINFO, Scopus, and Greenfile. With the help of a reference librarian, keywords related to children, school greening, physical activity, and socioemotional health were used. The search was conducted on 20 July 2020. Search terms included “greening schoolyards”, “green schoolyards”, “experimental design”, “quasi experimental design”, “green playgrounds”, “greening playgrounds”, “physical activity”, “physical exercise”, “children”, “kids”, “social health”, and “emotional health.” The full search strategy is attached in an Appendix A. The search results were exported to the Rayyan online tool for systematic review to facilitate the article selection process and ease the collaboration between the reviewers [41].

### 2.2. Article Selection Process

We applied a specific PICOS [42] criteria for inclusion and exclusion of studies in the systematic review as summarized below:

P (Population): Children under 18

I (Intervention): Greening the schoolyard or daycare yard and exposure of children pre/post-greening to the area.

C (Comparison): All types of control (such as other schools without green schoolyards) or self-controls (such as investigation of changes over time within same subjects in schools or daycare undergoing the interventions). The key comparison is between pre- and post-greening intervention.

O (Outcomes): Physical activity and/or socioemotional health using various measures for each outcome

S (Study design): Only experimental and/or quasi-experimental studies, all other design excluded.

Two reviewers independently screened titles of all results of the literature search against eligibility criteria, excluding those that clearly did not meet the decided PICOS criteria. Conflicts were resolved before repeating the same process, this time reviewing abstracts of all remaining articles. Next, the reference lists of all included articles were screened to identify relevant publications not retrieved by the electronic database searches. The aforementioned article selection process was repeated with the additional articles. The full methods section of each article was independently read by both reviewers to make sure no intervention other than children’s exposure to green space was included or missed in the review.

### 2.3. Eligibility Criteria

Inclusion criteria for the original search required the presence of keywords related to schoolyard greening in the title, abstract, or full text. This was done intentionally to capture the full scope of articles related to the topic.

Exclusion criteria were based on study design, intervention, and outcomes. Nonexperimental studies did not meet the inclusion criteria. Any study that did not have a schoolyard greening intervention or children’s exposure to green schoolyards was excluded. Reviews and protocols were excluded. Finally, any study that did not have physical activity or socioemotional health as an outcome was excluded.

### 2.4. Articles Selection Flowchart

The following flowchart (Figure 1) is a graphical illustration of PRISMA guidelines in the selection process for articles included in the review: 

## 3. Results

1843 records were identified by the original search (Ovid Medline: 449, PsycINFO: 210, Scopus: 1117, and Greenfile: 67). After removing duplicates, 1571 potential eligible articles remained. After title screening, 67 articles remained. After abstract screening, 21 articles remained whose full texts were assessed for eligibility according to the inclusion and exclusion criteria. References of included articles were checked for additional articles that might have been missed through the systematic database search. This process resulted in 6 articles that were included in this systematic review, as summarized in Table 1:

### 3.1. Interventions

Four of the six studies included in this review had used traditional schoolyard greening interventions [15,37,38,46] whereby the outdoor school environment is changed with a combination of natural elements (e.g., trees, flowers, sand, water, grass, hills, and bushes [37]) to create a more appealing schoolyard and improve the quality of children’s play experiences [37,48]. Two studies used modified versions of the interventions [43,45]. In a study by Hamer et al, the greening intervention consisted of adding AstroTfurf in schools’ major playground reconstructions [43]. Chiumento et al. used a recognized green space intervention called “Social and Therapeutic Horticulture (STH)” conducted with active engagement of children experiencing behavioral, emotional, and social difficulties [45]. The STH sessions were participatory, and children were actively engaged in the development of selected green spaces at each school [45]. With the children, they conducted six two-hour-long monthly sessions facilitated by two horticulturists and one psychotherapist to improve physical, mental, and social well-being as well as health equity [45].

### 3.2. Geographical Settings

The six included studies were distributed across two continents: two studies were conducted in North America and four studies were conducted in Europe. Of the two North American studies, one took place in the US [38] and the other in Canada [46]. Of the four European studies, one took place in Austria [15], two in the UK [43,45], and one in the Netherlands [37]. An Australian study was found and it was focused on the role of park greening on children’s physical activity; but it was ultimately excluded because it was not looking at schoolyard greening, which is the focus of this review [49]. No study was found in Africa, Asia, or South America; if they were found, the text may not have been in English and was therefore excluded from this review.

### 3.3. Participant Demographics

Two UK studies [43,45] and one US study [38] were conducted within inner city and low-income contexts with disadvantaged school children. The study in the Netherlands was conducted in moderate-to-urbanized areas with children in urban schools with limited green play opportunities [37]. The study in Canada did not report the urbanicity setting, although it was done in with childcare centers in Vancouver [46], and one study in Austria was conducted in a rural setting [15].

Per the inclusion criteria, all of the studies focused on children. The children’s age range varied from as low as 2 years old [46] to a high of 15 years old [15,45]. The sample size varied from as low as 36 children [45] to as high as 437 students [38]. The study by Chiumento et al. in 2018 used a sample of 36 children (9–14 years) in West England, UK [45], while Brussoni et al. in 2017 used a sample of 45 daycare children (2–5 years) in Vancouver, Canada [46]. Kelz et al. in 2015 conducted their study with 133 middle school students aged 13–15 years old (mean = 14.4 years) in three middle schools in a rural area in Gleisdorf, Austria [15]. Hamer et al. conducted their study in London, UK with a sample size of 231 (mean age = 8) students who completed follow-up (*n* = 169 intervention; *n* = 62 control) [43]. The study by van Dijk-Wesselius et al. in 2018 was carried out in the Netherlands (no city reported) with different sample sizes at different data collection points (238 at baseline, 233 at year 1 follow, up and 201 at year 2 follow up) aged 7–11 [37]. Raney et al. completed their study in Los Angeles, CA with a sample of elementary school children of 437: 82 controls and 355 experimental (no age range reported) [38].

### 3.4. Study Designs

All six studies included in this review used quasi-experimental design with pre-post data collection [15,37,38,43,45,46]. Four of the studies had only two pre-post data collection points [15,43,45,46], while the other two used three data collection points to assess longitudinal impact after exposure [37,38]. Two studies that use three data collection points both had comparison control schools that were not greened [37,38] in addition to the investigation of changes over time within same subjects in schools undergoing the greening interventions. Raney et al. collected data at baseline, immediately post-greening, and at 4 months after greening [38], while van Dijk-Wesselius et al. collected baseline data with follow-up at one year and two years post-greening [37]. In the four studies that used pre-post study design with two data collection points, two had comparison or control schools that did not receive the greening intervention [15,43], while the other two [45,46] compared their baseline pre-greening data to the post-greening data for impact assessment.

### 3.5. Outcome Variables

Outcomes measures varied somewhat across studies, although they all measured PA and/or SEH in some way. Two of the studies looked exclusively at SEH outcomes [15,45], while the other four measured both SEH and PA outcomes [37,38,43,46].

In the article selection process, there were four articles that met the outcome measures for inclusion but were excluded because they used case control study designs [50,51,52,53]. Those studies had no pre-greening data to compare to post-greening data. They instead compared greened schools to controls matched on school and neighborhood characteristics [50], school size or enrollment [52], and school sociodemographic characteristics such as the percentage of students receiving free or reduced lunches, students’ race/ethnicity, or school size [51]. One study did a field experiment to assess students’ physical activity differences between greened and nongreened areas of the school yards [53], then used the Personal Activity and Location Measurement System (PALMS) [54] to match the accelerometers and GPS data and calculate wear time and PA intensities. 

### 3.6. Data Collection Tools

The data collection tools varied across studies. In the four studies that looked at PA outcomes, three studies used only accelerometry data [37,43,46], while one study [38] combined accelerometry data and other observational tools, the System for Observing Play and Leisure Activity in Youth (SOPLAY) and the System for Observing Children’s Activity and Relationships during Play (SOCARP). SOPLAY is a validated tool for directly observing physical activity and associated environmental characteristics in free play settings [55], while SOCARP is a validated tool used to observe and to record child’s activity level, activity type, and social group size [56].

For studies using accelerometry data, the duration of the data collection varied. Brussoni et al. and Raney et al. collected five consecutive days’ worth of accelerometry data [38,46], while Hamer et al. had the students wear accelerometers for seven consecutive days during waking hours [43]. The study by van Dijk-Wesselius et al. limited the data collection to only one day a year at each school for the three data collection points [37]. Raney et al. collected SOPLAY and SOCARP data and did not report the number of days they collected the data, nor the time of day during which students were observed [38].

For the socioemotional health (SEH) outcome, a variety of tools were used to capture a variety of constructs. In the assessment of social behaviors and interactions, van Dijk-Wesselius et al. and Brussoni et al. videotaped students during play sessions [37,46]. Brussoni et al. video results will be discussed in the findings section, while van Dijk-Wesseluis et al. video results were still being analyzed and were not included in their study [37], and therefore not included in this review. Raney et al. used SOCARP to record a target child’s social group size and social interactions [38]. In addition to the videotaped data, Brussoni et al. also used the Strengths and Difficulties Questionnaire (SDQ) teacher version [57] and the Preschool Social Behavior Scale Teacher Form (PSBS-T) [58] questionnaires to assess social behaviors [46], while van Dijk-Wesselius et al. used the Social Orientation Choice Card (SOCC) [59] to asses children’s prosocial orientation [37].

The studies used varying tools to measure mental well-being. Kelz et al. used two well-being indices: (1) the intrapsychic balance subscale of the standardized Basler Well-Being Questionnaire [60] and (2) the Recovery-Stress Questionnaire (R-SQ) [61] to assess pupils’ recovery from stress [15]. Chiumento et al. used the 7-item Warwick–Edinburgh Mental Well-being Scale [62,63] and the Mental Well-being Impact Assessment (MWIA), an evidence-based qualitative tool which aims to assess the potential impact of a specific policy, service, project, or program on the mental well-being of a population in three domains: increasing resilience and community assets, enhancing control, and participation and social inclusion [45]. van Dijk-Wesselius used the emotional functioning subscale of the Dutch Pediatric Quality of Life scale [64,65] which is a measure of emotional well-being [37].

The restorative quality of greened schoolyards was also assessed by different instruments. For example, Kelz et al. used the conflict scale of the Attention Network Test (ANT), a measure of executive functioning [66] and the Perceived Restorativeness Scale (PRS) [67] to determine the subjective impression of the restorative qualities of the schoolyard before and after a greening intervention [15]. van Dijk-Wesselius et al. used the Perceived Restorative Components Scale for Children [68] to measure children’s perception of the restorative quality of the schoolyard [37] as well as two attentional tests: the Digit Letter Substitution Test (DLST), a measure of information processing speed [69], and the Sky Search task (SST), a subscale of the Test of Everyday Attention for Children [70], a measure of selective attention to assess attention restoration before and after recess periods in green or nongreen environments [37].

### 3.7. Measures of Outcome Variables

The measures of outcome variables varied across studies. For the PA outcome, van Dijk-Wesselius et al. reported the percentage of time spent in moderate to vigorous physical activity (MVPA) [37], while Brussoni et al. reported the total time spent in MVPA [46]. Hamer et al. reported the total time spent in sedentary PA, in light PA, and in MVPA [43], while Raney et al. reported mean percentages of time spent in sedentary PA, in light PA, in MVPA, and percentage of number of children observed in sedentary PA and in MVPA [38]. Raney at al. also reported zone popularity before and after the intervention, in addition to percentages of the number of children observed in different activity intensity and mean percentages of time spent in different activity intensity [38].

In reporting measures of SEH outcomes, authors reported on similar constructs using a variety of units of measure. Raney et al. reported the change in physical and verbal conflict rates as measures of prosocial or antisocial interactions [38], while van Dijk-Wesselius et al. reported the percentage of children engaged in prosocial behavior from the Social Orientation Choice Card game [37]. Brussoni et al. reported scores from the Strengths and Difficulties Questionnaire (SDQ) teacher version, scores from the Preschool Social Behavior Scale Teacher Form (PSBS-T), and the seven C scores [71] as measures of mental well-being and emotionally positive interactions between children. Higher SDQ scores are associated with greater likelihood of a psychiatric diagnosis [46]. Brussoni et al. also reported odds ratio for different measures of children’s engagement in pre-coded play specific behaviors, including prosocial behaviors, antisocial behaviors, and solitary play [46].

Three studies [15,37,45] reported mental well-being using different instruments. Chiumento et al. reported the Well-being check cards scores as well as qualitative findings from the Mental Well-being Impact Assessment [45]. Kelz et al. reported the mean scores for indicators of well-being using the intrapsychic balance (BBS) score and overall well-being score [15], while van Dijk-Wesselius et al. reported the emotional well-being score from the Pediatric Quality of Life scale [37]. Brussoni et al. reported mean scores for the SDQ test and seven C scores as indicators of mental and emotional well-being [46].

Three studies reported the student’s perception of their greened schoolyard [15,37,43]. Kelz et al. reported multiple indicators including the mean scores of four different indicators for perceived restorativeness (being away, fascination, coherence, and compatibility) and the mean conflict score measure of executive functioning [15], while van Dijk-Wesselius et al. reported the mean scores for four indicators of subjective schoolyard perception (naturalness, likability, attractiveness, and restorative quality) [37]. Hamer et al. assessed the perceived positive change post-greening in well-being and social interactions through qualitative interviews [43].

### 3.8. Analytical Approaches

The analytical approaches used to determine the effects of the interventions on different outcome variables varied depending on the study design and the type of data. All studies used some type of quantitative analysis, and three studies incorporated some qualitative analysis such as interviews or focus groups [43,45,46]. Five of the six studies used regression analyses [15,37,38,43,46] to test for the statistical difference between children’s PA and SEH outcomes at different time periods before and after the greening intervention. In one study, Chiumento et al. used mean score comparisons in the analysis of quantitative well-being check cards [45]. One study by Raney et al. incorporated correlation analysis to compare the relationship between physical activity level and social interactions [38], while Brussoni et al. incorporated the nonparametric Wilcoxon Signed Rank tests to compare statistical differences between related samples [46].

## 4. Discussion

### 4.1. Impact of Greening on Children’s Physical Activity

While the studies differ in methodological approaches, participants’ age ranges, data collection techniques, and sample sizes, the studies confirmed some positive effects of green schoolyards on children’s PA, in at least a subgroup of their sample, with some exceptions on PA outcome measures. The findings related to PA are summarized in Figure 2, examining each comparison made within each study according to the specific outcome measures used and the overall direction and significance of the findings.

Schoolyard greening was associated with a beneficial change in children’s PA in many, although not all, PA outcome measures. Raney at al. and Hamer at al. reported no significant change in overall time spent in light PA post-greening [38,43], though Hamer at al. found an increase of time spent in light PA for students younger than nine [43]. Hamer et al. also found no significant differences in time spent in either moderate or vigorous PA post-greening; however, they did find a positive association between schoolyard greening and a decrease in time spent sedentary for children younger than nine [43]. Of note, the study by Hamer et al. examined an intervention to install AstroTurf only. One study by Brussoni et al. was focused on promoting nature and risky play behaviors by adding natural materials to the playground [46]. Brussoni et al. found a decrease in total time spent in MVPA post intervention [46].

Findings on remaining PA outcomes (zone popularity, time spent in MVPA, percentage of children observed in sedentary PA, and percentage of children observed in MVPA) were beneficial for children’s PA with some age and sex specific differences. van Dijk-Wesselius et al. found a positive effect of greening on girls’ PA while no impact was found in boys [37]. Similarly, Raney et al. found no greening impact on boys’ PA and a positive impact on girls’ PA and 5th graders’ PA [38]. Raney at al. also looked at schoolyard utilization and found positive associations between schoolyard greening and increase in utilization of greened areas in the schoolyard [38], as measured by zone popularity. Zone popularity was similar between the experimental and control students at baseline while the number of students significantly increased in areas replaced by green space, an indicator of the impact of greening on area use.

Two studies found that schoolyard greening significantly stimulates physical activity in girls by increasing the amount of time they spent in MVPA during recess, particularly shortly after greening [37,38]. This may be an indication that schoolyard greening provides children with a better play environment and increases their play opportunities, particularly for girls whose degree of change after greening was 12.2%, compared to boys at 6.2% [38]. These findings suggest that greening of the schoolyard can help in reducing physical activity equity gaps [72], such that females may have had limited play opportunities compared to their male counterparts who typically engage in competitive sports on hardscapes [38] and therefore engage in more PA on hard surfaces [73] and spend less time sedentary during recess on hardscapes [74,75]. Raney et al. suggested that the positive change observed shortly after greening might have been caused by more enjoyable opportunities for creative free play, which reduces boredom and increases students’ motivation for playing [38]. Similarly, van Dijk-Wesselius et al. found a high magnitude of change for girls at the first year follow-up, a change which declined at the second year follow-up, although their study was limited by collecting data for a single day per year [37].

### 4.2. Greening Impact of Children’s Socioemotional Health

Findings related to socioemotional outcomes varied across studies as illustrated in Figure 3, which shows comparisons made by each study and the outcomes and overall findings of each comparison.

For many outcome measures investigated, there was a beneficial change in children’s SEH because of greening, though there were some that found no impact of certain measures, and in some subgroups, a negative impact was noted. Three studies [37,38,46] found a positive or mixed association between schoolyard greening and some children’s prosocial behaviors, though they used different measures. Raney et al. observed a significant decrease in physical and verbal conflicts after greening [38], while van Dijk-Wesselius el al. observed a positive effect on younger children’s social functioning, particularly on social support and self-reported peer problems, and a negative effect was observed on self-reported prosocial behavior for older children [37]. van Dijk-Wesselius also used Likert scales to assess children’s perception of the schoolyard’s naturalness, likability, attractiveness, and perceived restorative quality, finding that children showed greater appreciation of the greened schoolyard, particularly younger children and girls [37]. Brussoni et al. found that greening had a positive impact on emotionally positive social interactions [46]. Because the study by Brussoni et al. combined interventions, it is unclear whether changes observed are attributable to changes to the environment or to programming offered.

Two studies [15,37] found a positive association between schoolyard greening and attention restoration. Kelz et al. found a positive effect for two subscales of the Perceived Restorativeness Scale (compatibility and fascination), while one of the subscales (being away) did not change and another subscale (coherence) decreased post-greening [15]. Additionally, no impact was observed for another measure, executive functioning, although the study authors noted that prior to the intervention, executive functioning was significantly higher in the experimental school than the control school, which might have influenced the findings, leading to an underestimation of the greening effect on executive functioning [15]. van Dijk-Wesselius used attention restoration tests (SST and DLST) and found that children’s scores on the two attentional tasks improved after recess in greened schools at the second follow-up, suggesting an impact after the schoolyard had already been greened for a longer period [37].

Three studies found a positive association between schoolyard greening and mental well-being [15,45,46]. One study found no greening impact on children’s emotional well-being; however, in that particular study, there were few to no emotional problems among the children at baseline, which could explain why the intervention had minimal impact on the measure [37]. Additionally, the quality and quantity of the green space might have influenced the results leading to an underestimation of the greening impact on emotional well-being [37]. Kelz et al. observed a significant increase in the intrapsychic balance scores for children in the schoolyards after greening compared to times of measurement at the control school and the first time of measurement at the experimental school [15]. Brussoni et al. found a significant decrease in the SDQ peer problems scale and the PSBS depression score after the intervention [46]. Chiumento et al. found a positive impact of the intervention on the MWIA, while the well-being card scores showed many negative effects, although they were not found to be statistically significant [45]. Because the study by Chiumento et al. combined interventions, it is unclear whether changes observed are attributable to changes to the environment or to programming offered.

## 5. Conclusions

This review sought to explore the relationship between schoolyard greening and children’s well-being, particularly levels of physical activity and socioemotional health. Many outcome measures for both PA and SEH showed beneficial changes due to greening, with some exceptions (Figure 2 and Figure 3).

In regard to socioemotional health outcomes, all authors reported beneficial effects of greening with only a few exceptions (Figure 3). For instance, van Dijk-Wesselius found that greening had positive impact on many outcome measures, except the measures of children’s emotional well-being, children’s perceived restorative quality of the greened schoolyards, and the negative effect on older children’s prosocial behaviors [37]. It is important to note that van Dijk-Wesselius et al. also found on two attention restoration tests, the SST and the DLST, that greening does seem to promote restoration by replenishing depleted cognitive resources post-recess [37]. The effect was observed at the second time point [37], suggesting that beneficial effects of greening on attention restoration are observed longer after greening. Kelz et al. found that greening has positive effects on measures of SEH with the exception of measures of executive functioning and two subscales of the Perceived Restorativeness Scale (being away and coherence) [15]. Chiumento et al. reported positive impact in their qualitative portion; however, they found a negative effect in many domains of the well-being check cards, though they were not statistically significant [45]. This heterogeneity in the findings suggests the need for more robust experimental evidence regarding the impact of greening on physical activity and socioemotional health outcomes.

The findings from this review have implications for guiding future research directions, informing policy and decision-making processes, and promoting health equity. Future studies on the impact of schoolyard greening on children’s PA and SEH should consider more rigorous approaches in methodology, data collection, and analysis to limit sources of error. To do this, researchers should consider including multiple days of data collection and combining observational tools such as SOPLAY and SOCARP with GPS and accelerometer data to accurately assess all the schoolyard factors that contribute to any associations. Researchers should consider using quasi-experimental studies with comparison groups and observe student cohorts prospectively. Longitudinal studies investigating the health benefits of early and equitable access to green schoolyards would provide further evidence on the role of green spaces in promoting lifelong health in communities. The new knowledge generated from such research will support policy makers such as school boards, education departments, nonprofit organizations, and other stakeholders in making evidence-based health policy decisions.

Finally, schoolyard greening may offer an opportunity to reduce health equity gaps by improving school children’s physical and mental health. Early exposure to green space through access to greened schoolyards is a promising venue for reducing health inequities between children in low-income urban neighborhoods with limited access to green space and those growing up in high-income neighborhoods with abundant and high-quality green space. Additionally, greening promotes equal play opportunities for male and female students alike. Greening low-income schoolyards which are often covered in asphalt or concrete would ensure equitable use of those greener and healthier schoolyards by all children regardless of their location or socioeconomic status. This equitable use of those healthier schoolyards would result in beneficial outcomes for all children regardless of their race, ethnicity, age, residential neighborhoods, or parental socioeconomic status, therefore enhancing health equity through accessibility and exposure to healthier and greener schoolyards. 

## Figures and Tables

**Figure 1 ijerph-18-00535-f001:**
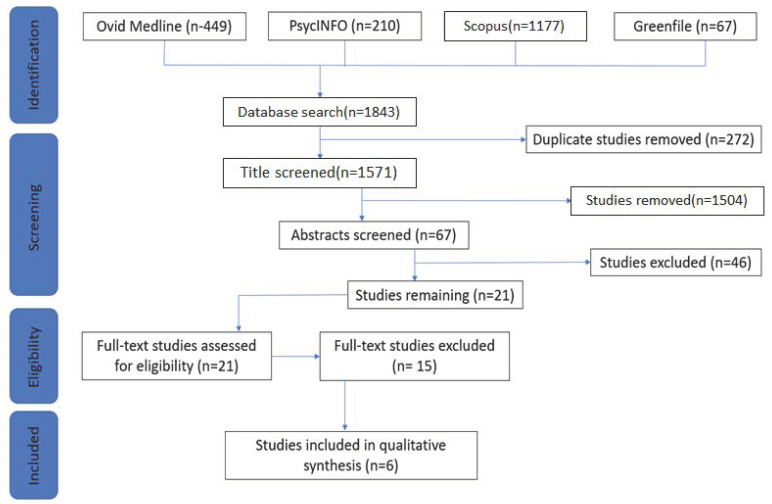
Graphical illustration of PRISMA guidelines in the articles’ selection process.

**Figure 2 ijerph-18-00535-f002:**
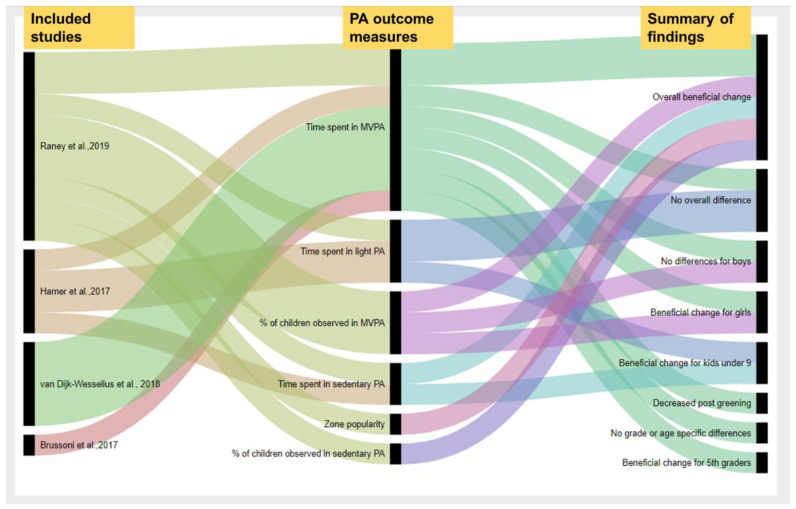
The impact of schoolyard greening on physical activity outcomes: The first column represents the authors, the second column represents the physical activity (PA) outcome measures, while the third column represents the summary of findings for each outcome. This graphical representation shows an overall trend in findings across studies.

**Figure 3 ijerph-18-00535-f003:**
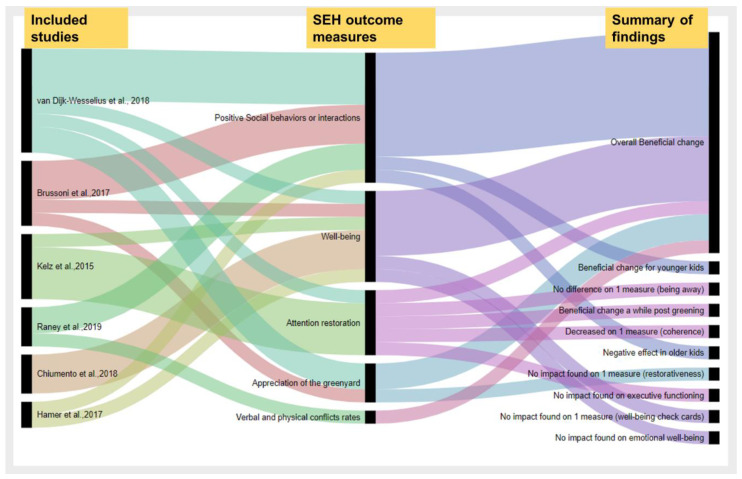
The impact of schoolyard greening on socioemotional health outcomes: The first column represents the authors, the second column represents the SEH outcome measures, while the third column represents the summary of findings for each outcome. This graphical representation shows an overall trend in findings across studies.

**Table 1 ijerph-18-00535-t001:** Main characteristics and results of included Studies.

Citation/Setting	Study Design/Data Sources	Variables	Analytical Approach	Results/Conclusions	Strengths and Limitations
van Dijk-Wesselius et al. (2018) [37] The Netherlands (moderate to urban)	**Design:** Longitudinal Prospective, measurements at baseline, 1 year, and 2 years in 9 schools: 5 experimental and 4 controls matched on level of urbanization of the neighborhood and socioeconomic status of parents. **Sample size** with boys percentages at intervention and control respectively: **Baseline**: *n* = 706 (48.6%; 52%); **Year 1**: *n* = 683 (44.7%; 52.2%), **Year 2**: *n* = 643 (48%;56.1%). Children ages 7–11 in urban schools with limited green play opportunities. **Data source**: Video observations of children’s play behavior and accelerometer-based physical activity measured at the schoolyard, classroom-based tests of children’s attentional capacity and social value orientation. **Self-reported data** to assess children’s perception of schoolyards attractiveness and their social and emotional well-being Interviews with school principals and questionnaires amongst teachers and parents of the schools.	**Independent variable:** Exposure to green school yards **Outcomes:** **Physical activity:** Measured by % of time spent in MVPA. **Social behavior:** Prosocial orientation % Prosocial Peer problem Social support **Perception of schoolyards:** (naturalness, likability, attractiveness, and perceived restoration). **Emotional well-being:** Measured by the subscale emotional functioning of the Pediatric Quality of life scale. **Attention:** **DLST** (Measure of information processing speed) **SST** (Measure of selective attention) **Covariates:** Parental SES and education level, gender, and group of children. **Hypothesis:** Greened school yards children would display (1) more positive appreciation of the schoolyard (2) increased levels of physical activity during recess (3) more attention restoration after recess, and (4), improved (pro)social behavior and (5) better emotional functioning.	MLwiN software for Multilevel data analysis to control for the (partial) clustering of measurements within children (repeated measures) and the clustering of children within schools. First, intercept-only models were fitted with separate random intercepts for the three times of measurements at child and at school level. 2nd, models with covariates. 3rd, models to estimate the main effect of time. 4th, the main effects of school conditions (intervention vs. control) Last, models to estimate the effects of greening of the schoolyards at first and second follow-up	Greening has a positive impact on children’s appreciation of the schoolyard, particularly in younger students and girls. Greening has a positive impact on children’s attention restoration after recess but only after the schoolyard had been greened for a longer period. Greening has a positive short-term impact on younger children and a negative impact on older children’s prosocial orientation. Greening is beneficial for children’s social functioning, particularly on social support and self-reported peer problems but not on self-reported prosocial behavior. Greening stimulates physical activity, but only in girls particularly shortly after the schoolyard has been greened. No impact was found on children’s emotional well-being.	**Strengths:** Many theories/models: Biophilia hypothesis, Stress recovery theory, Attention restoration theory, theory of loose parts, affordance theory. Use of several validated scales, using matched controls and following up more than once. **Limitations:** **The quasi-experimental** design nature might have led to the selection bias. Impossible to randomize schools to experimental or control. The study only included schools from **moderate to highly urban areas**. It can’t be generalized to children in more rural and green areas. **Quantity and quality** of greening could have led to an underestimation of the impact because all greened schools still had some paved areas. **The study** doesn’t allow any conclusion about the impact of greening schoolyards on children’s individual development over time because of using a “between subjects design”. To answer that question, a “within subjects design” study would be needed. Data collection was also **limited to only one day** a year at each school for the three data collection points.
Kelz et al. (2015) [15] Gleisdorf, Austria (Rural) 51% male from experimental 52% male from controls schools All 4th graders aged 13 to 15 years old.	**Design:** Pre–post test quasi experimental field research in a natural context: renovation of a secondary school with students up to age 18. Compare data to two nearby schools over the same time period **Sample size:** 133 middle school pupils with average age of 14.4 years old **Data source:** Physiological stress was assessed by blood pressure measured by devices from BOSO **ATN:** The Attention Network Test used to assess executive functioning by three scores: alerting score, the orienting score, and the conflict score. **Well-being** was assessed by two indices: (1) the intrapsychic balance subscale of the standardized Basler Well-Being Questionnaire and (2) the Recovery-Stress Questionnaire (R-SQ) to assess pupils’ recovery from stress. **Perceived Restorativeness:** assessed by the Perceived Restorativeness Scale (PRS) to determine the subjective impression of the restorative qualities of the schoolyard before and after the renovation.	**Independent variables:** Exposure to green schoolyards **Outcome variables:** Executive functioning Psychological well-being, and Perception on the restorative nature of green environment. **Covariates:** Demographics variables such as: parents’ level of education, amount of greenery near the kid’s home, time spent outdoors, and sports involvement. Weather was controlled for because schools are on the exact same streets (therefore same weather). **Hypothesis:** Exposure to green schoolyards has beneficial effects on humans’ restoration, reduction of physiological stress (blood pressure and heart rate), enhancement of cognitive executive functioning, improvement of psychological well-being, and a general perception of green school yard being more restorative compared to non-green schoolyards	Multi methods approach: The first 3 hypothesis were tested by planned comparisons tests to contrast the experimental school’s pupils’ mean of the second measurement (after schoolyard installation) against the mean of the other three means (both measurements of control school’s pupils and first measurement of experimental school’s pupils [15]. The 4th hypothesis was tested by t-test for paired samples	The physiological stress indicators were lower for the experimental school’s pupils at the second time of measurement compared with the mean of both times of the control school’s measurements and the experimental school’s first time of measurement. There was **no greening effect on executive functioning** as opposed to what was hypothesized. **Well-being improved post-greening:** the intrapsychic balance scores for kids in the greened schoolyards were higher after greening compared with the mean of both times of measurement at the control school and the first time of measurement at the experimental school. For the two PRS subscales (compatibility and fascination), **perceived restoration increased** pre- to post-renovation in the interventional school. Being away showed no changed while coherence decreased **Recommendations to close the gap:** Only ages of 13–15 were studied. Further studies would look at a broader range of ages: 6–18. Impact would be higher in studies where kids have no access to nature (urban areas). Impact on academic performance also needs to be investigated. Kids behavior at home also could be assessed by parental reports.	**Strengths:** Longitudinal pre post Quasi-experimental design nature of the study rules out confounding demographic differences. **Many theories:** psycho-evolutionary theory, biophilia theory, attention restoration theory, **Referenced many other studies:** (1) Boston schoolyard initiative in which half schools was greened and the other half wasn’t and (2) the Chicago public housing project. **Consistency** with previous findings remain the same: Improvement on increased learning opportunities, academic performance, better physical activity and mental health **Limitations:** Experimental and control schools were different in type (secondary vs. secondary modern school type) in kids admission process. Time difference between pretest (March) and posttest (June). Results might have been influenced by seasonal climatic and school related influences. Financial barrier led to lower degree of greenery in the experimental school.
Raney et. al. (2019) [38] Los Angeles, CA, USA: Urban Low income schools (5-day consecutive wear and excluding from analysis kids with <3 days of accelerometer wear.	**Design:** Pre post experimental design: Stepwise impact evaluation study **Sample size: (*n* = 437)** Experimental kids: 355 vs. control kids: 82 **Data source:** Scanning zones during recess using SOPLAY and SOCARP to record a target child’s activity level, social group size, activity type, and social interactions. 3 accelerometers data collection points: pre-, immediately post-, and 4 months post-greening	**Independent variables:** Exposure to green space **Outcome variables:** (1) **Physical activity level** (sedentary, walking/moderate, or vigorous) (2) **Recess specific behavior** or social behavior (social group size, activity type, and social interactions). **Hypothesis:** green space increase daily actively level among children and increase better social behavior	**Linear mixed models** Determining main and interaction effects of condition, study phase, sex, grade, and surface type on play behaviors. **Pearson correlation:** Analysis of relationships between activity levels and social interactions such as group size, time spent in pro social vs. antisocial activity.	“Zone popularity and recess behaviors did not change for control students during the study; but changed for experimental students.” “Green space exposes children to nature and increases daily MVPA levels and promote social well-being in sex and age dependent ways.” Physical and verbal conflicts decreased post-greening over time. It takes kids some time to adapt to new changes [38]. **Recommendation:** There are no studies looking at sex specific and age group specific PA levels in green schoolyards, therefore they are needed.	**Strengths:** Using a control and multiple data collection techniques for specific analysis, longitudinal nature of the study and multiple simultaneous comparisons between hardscape vs. green space, sexes and grades. **Limitations:** Single experimental place—leads to no generalization to different population demographics. Supervisors limiting student movement to asphalts due to safety concern. No data related to yard supervision-student interactions were collected.
Hamer at al. (2017) [43] London, UK Deprived inner city areas. Small greening aspect in the intervention (addition of the new AstroTurf to the playground). 7 days accelerometry data. At least 1 school day and 500 min of measured monitor wear time for analysis.	**Design** Quasi-experimental: comparison group pre-test/post-test design). **Sample size: (*n* = 231):** Experimental: 169 (53.2% Male) and control: 62 (46.8% Male) Mean age: 8 **Data source:** Accelerometers worn during waking hours but not during water-based activities or sleep. Body composition analyzers and height measurements	**Independent Variables** Access to green schoolyard. **Outcome Variables:** **Physical Activity** measured by accelerometers following the International Children’s Accelerometry Database Study protocol [44]. **Sedentary:** <100 counts per minute (cpm), Light activity: 100 to 3000 cpm, and moderate to vigorous physical activity **(MVPA):** >3000 cpm. **Body mass index (BMI)** Derived from weight and height measurements **Covariates:** Age, sex and ethnic background	**Independent Sample t-test** to text for differences in baseline data **Mixed models** to compare PA at follow up between intervention and controls **Thematic analysis** for analysis of qualitative focus group and interview data	After adjusting for multiple factors, no overall difference was observed in light activity, moderate-vigorous activity, or sedentary time. There were age interactions for sedentary and light activity, with reduction in total sedentary time and increase in light intensity activity for children <9. “Qualitative data suggested that the children enjoyed the new playgrounds and experienced a perceived positive change in well-being and social interactions”.	**Strengths:** Quasi experimental study design and control of major confounding variables Assessment of interventions by independent researchers **Weaknesses:** No consensus on cut points in kids’ accelerometry studies. Conservative cuts points were used. No randomization of schools
Chiumento et al. (2018) [45] North West England Kids experiencing behavioral, emotional and social difficulties.	**Design:** Mixed-methods pre–post evaluation design. **Sample size (*n* = 36)** School children aged 9–15 years. **Data Source:** The Mental Wellbeing Impact Assessment (MWIA) qualitative tool and the and the 7-item Warwick-Edinburgh Mental Well-being Scale were administered pre–post intervention.	**Independent variable** Exposure to the Social and Therapeutic Horticulture (STH) green space intervention. **Outcome** Mental well-being Measure with the Mental Well Being Impact Assessment (MWIA) and Wellbeing Check Cards	**Qualitative analysis** used for the MWIA results **Quantitative analysis** used to compare Scores from the Wellbeing check cards pre- and post-intervention	Group based socially interactive horticulture activities facilitated by trained therapists are associated with positive impacts upon the mental and emotional well-being of children experiencing behavioral, emotional and social difficulties. Wellbeing check cards had worse scores in many domains post intervention, no results seemed to be statistically significant.	**Limitations:** The study was a pilot and had a small sample size
Brussoni et. al. (2017) [46] Vancouver, Canada Two childcare centers (A and B) Accelerometer worn for 5 days from arrival at the childcare center to departure back home	**Design:** Convergent mixed methods repeated measures design to examine the effect of the intervention with two-points data collection (pre–post) **Sample size:** (n = 45) [53% boys] Age:2–5 years (Mean age = 4.28 years; SD = 0.63). **Data source:** Two-point data collection (pre–post). The “Seven Cs” The Strengths and Difficulties Questionnaire (SDQ) teacher version and the Preschool Social Behavior Scale Teacher Form (PSBS-T). Accelerometry data Videotapes of the 30-min play sessions to assess social behaviors ECE focus groups	**Independent** Exposure to green space **Outcome:** **Seven Cs score:** Assessed by the 27 items on seven Cs (character, context, connectivity, clarity, change, chance, and challenge) rated on a 5-point scale, for a maximum score of 135, higher scores associated with more positive emotional interactions. **Social behavior:** Measured by the children’s SDQ and PSBS-T scores **Physical activity** was measured with validated accelerometers worn by 3–5 years old [47]. Sedentary:199 count/15 s Light: 200–419 count/15 s Moderate: 420–841 count/15 s Vigorous: >842 count/15 s. **Play activity and social interaction behaviors:** Observation data on non-rainy days with specific codes	**Wilcoxon Signed Rank tests:** used to compare Children’s scores on SDQ and PSBS-T tests before and after the intervention. **Generalized linear mixed effects models (GLMM):** To examine change in likelihood of engaging in coded behaviors from observations and videos. They used a random effect model adjusting for the length of video, time of day, and day of week **Qualitative data analysis:** for the two semi-structured focus groups with ECE staff at T2, one at each center, to assess perceptions on the play space prior to the intervention, changes to the play space and observations on changes in children’s behavior.	**Results** Seven Cs scores increased from 44 to 97 in Centre A, and 35 to 125 in Centre B There was a significant decrease in the SDQ peer problems scale from T1 (T1 median 2.3) to T2 (T2 median 2.0) z = −2.10, *p* = 0.036. **MVPA** decreased from T1 to T2 (M decrease 1.32min, SE = 0.37, *p* <0.001) **The PSBS Depression** score decreased significantly from T1 (T1 median 6.0) to T2 (T2 median 3.0) z = −2.24, *p* = 0.03. **The qualitative analysis** suggests positive perceptions of the school yard post the intervention. **GLMM showed mixed** results in the play observations from T1 to T2: *Risky play*—no increase *Play with natural materials* increased. *Prosocial behavior*: Girls and older children were more likely to engage in *prosocial behavior* than boys and younger children [46]. *Prosocial behaviors* increased in Center A (OR = 2.81) but decreased in Center B (OR = 0.17). *Antisocial behavior* did not change in Centre A, but decreased significantly in Centre B (OR = 0.16). *Engagement with play*: Boys were less engaged compared to girls (OR = 5.11). Kids in Center B were more engaged compared to kids in Center A (OR = 0.23) *Solitary play:* No change. Boys are more likely to engage in solitary play than girls.	**Limitations:** It is difficult to isolate intervention effects from typical development in longitudinal intervention research with young children T1 and T2 were in different seasons, effect of seasons is unknown (T1 was collected in winter, T2 in spring) SDQ data analysis violated the “symmetry of differences assumption” and may be spurious. **Strengths:** Intervention effects were estimated with odds ratio and 95% confidence intervals.

## Data Availability

No dataset was used in this study because it is a systematic review, therefore data is contained within the article.

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
