# Peer review of "The Impact of Schoolyard Greening on Children’s Physical Activity and Socioemotional Health: A Systematic Review of Experimental Studies"

_ijerph, 2021, doi:10.3390/ijerph18020535_

Round 1
Reviewer 1 Report
Your review of experimental studies of the impact of schoolyard greening on health outcome with respect to physical activity and socio-emotional health is thorough. At the end, you have provided some recommendations for further studies, which is great. Overall, you have done a good job.
Author Response
Thank you for the thoughtful and careful review of our manuscript. We think this is an important area for intervention to improve children’s health outcomes by focusing on places where they spend most of the daytime (schools).
Reviewer 2 Report
The paper works on an really interesting subject: green schools and their impact about socialization, wellbeing and future professional developments, but the authors present a merely descriptive paper. It is necessary that the authors voices strongly appear, not only what other researches say or have said about this topic.
Other important questions to resolve are: the paper played by culture in each one of the countries where the studies was made. It is a key factor? The authors say that greening school has a possitive effect in children's welbeing, but they recognize that poverty is related with urban scholarization and no greening schools. This means that greening schools can be a healthy solution but, at the same time, can be related with inequalities. Can they develop this idea?
If the authors develop this points i reccomend the journal acepts this paper
Author Response
Thank you for reviewing our manuscript and for your thoughtful and helpful comments. We have revised accordingly, and we believe the manuscript is much improved as a result. We have detailed our revisions in response to your comments below and have indicated revisions in the revised manuscript.
Reviewer 3 Report
Lines 85-88- Methods- If registered, provide the name of the registry (e.g., PROSPERO) and registration number.
Line 87- Explain why Prisma guidelines were used, instead of Prisma-P’ ones (2015).
Line 91- explain criteria used to select databases (and why Prospero was not included)
Lines 97-99- explain how duplicate detection, resolution and reporting was made
Line 101- explain why the use of PICOS instead of PICO
Line 112- Did you wanted to say experimental and/or quasi-experimental studies?
Table 1 (if possible):
- Identify gender distribution, of van Dijk-Wesselius et al. (2018) and Hamer at al. (2017) studies.
- Identify gender distribution, of Raney et al. (2019) study.
- Identify standard deviation of children' ages, of Hamer at al. (2017) study.
- clarify abbreviations (cpm, MVPA) and protocol followed, in Hamer at al. (2017) study.
- Identify gender distribution, of Chiumento et al. (2018) study.
- Identify gender distribution, of Brussoni et al. (2017) study. (and correct reference: et.)
- clarify protocol followed for physical activity, in Brussoni et al. (2017) study.
- In Brussoni et al. (2017) study, verify if effect size was estimated for Wilcoxon, if not it should be noticed as a limitation (as revealed “SDQ data analysis violated the “symmetry of differences assumption” and may be spurious.”), also verify confidence interval.
Lines 310-321- Analytical Approaches- Explain constraints that impeded meta-analysis.
Author Response
Thank you for reviewing our manuscript and for your thoughtful and helpful comments. We have revised accordingly, and we believe the manuscript is much improved as a result. We have detailed our revisions in response to your comments below and have indicated revisions in the revised manuscript.
Response to Reviewer 3 Comments
Point 1: Lines 85-88- Methods- If registered, provide the name of the registry (e.g., PROSPERO) and registration number.
Response 1: We have submitted a registration to PROSPERO. It is currently pending, and we do not have a registration number yet. Therefore, we did not report on this in the paper. We are happy to provide this registration number when it becomes available.
Point 2: Line 87- Explain why Prisma guidelines were used, instead of Prisma-P’ ones (2015).
Response 2: The software we used (Rayyan) is compatible to PRISMA guidelines for reporting and PRISMA guidelines are still well used in the literature, so we decided to use PRISMA. In the future, we will consider Prisma-P for our reviews.
Point 3: Line 91- explain criteria used to select databases (and why Prospero was not included)
Response 3: We needed to include all citation databases of peer-reviewed literature with potential of having articles focused on greening (GreenFile), source neutral abstract and citation database curated by independent subject matter experts (Scopus), medical sciences from the National Library of Medicine's bibliographic database (Ovid Medline) and a pool of citations and other abstracts of literature in the field of psychology (PsycINFO). With those four databases, we felt we would capture all relevant content in addition to the articles’ respective reference lists. We did not use PROSPERO because systematic reviews were excluded in our inclusion criteria.
Point 4: Lines 97-99- explain how duplicate detection, resolution and reporting was made
Response 4: The online software we used (Rayyan) in the article selection process helped us identify all duplicates. We deleted duplicate records and kept only one copy of the citation in the software.
Point 5: Line 101- explain why the use of PICOS instead of PICO
Response 5: We used PICOS instead of PICO because part of our inclusion criteria was based on specific study designs. Only experimental and/or quasi-experimental studies were included in our review.
Point 6: Line 112- Did you wanted to say experimental and/or quasi-experimental studies?
Response 6: We meant to say experimental/and/or quasi experimental studies. We are updated this in the paper. It is now on Line 117.
Point 7: Table 1 (if possible):
Point 7 (a) - Identify gender distribution, of van Dijk-Wesselius et al. (2018) and Hamer at al. (2017) studies.
Response 7 (a): Thank you for this important comment. We have addressed this point as suggested in Table 1. For Dijk-Wesselius et al. (2018), we included the gender distribution for each data collection point as it can be seen in our table 1. For Hamer et al. (2017) with experimental kids: n = 169 (53.2% Male) and control kids: n = 62 (46.8% Male).
Point 7 (b) - Identify gender distribution, of Raney et al. (2019) study.
Response 7 (b): There is no gender distribution reported.
Point 7 (c) - Identify standard deviation of children' ages, of Hamer at al. (2017) study.
Response 7 (c): No age SD was reported in Hamer et al. study. They only reported mean age, which we have in Table 1.
Point 7 (d) - clarify abbreviations (cpm, MVPA) and protocol followed, in Hamer at al. (2017) study.
Response 7 (d): We defined cpm (count per minute) and MVPA (moderate to vigorous physical activity). We also added the protocol used in measuring physical activity in Table 1: International Children’s Accelerometery Database Study protocol.
Point 7 (e) - Identify gender distribution, of Chiumento et al. (2018) study.
Response 7 (e): There is no gender distribution reported by Chiumento et al. study.
Point 7 (f) - Identify gender distribution, of Brussoni et al. (2017) study (and correct reference: et.)
Response 7 (f): We corrected references and added the gender distribution, mean age and standard deviation in Table 1.
Point 7 (g) - clarify protocol followed for physical activity, in Brussoni et al. (2017) study.
Response 7 (g): We included the protocol used in measuring physical activity in Table 1. (validated accelerometers worn by 3-5 years old).
Point 7 (h) - In Brussoni et al. (2017) study, verify if effect size was estimated for Wilcoxon, if not it should be noticed as a limitation (as revealed “SDQ data analysis violated the “symmetry of differences assumption” and may be spurious.”), also verify confidence interval.
Response 7 (h): In Brussoni et al. (2017)’s table 1, they reported interventions effects with odds ratio and 95% confidence intervals.
Point 8: Lines 310-321- Analytical Approaches - Explain constraints that impeded meta-analysis.
Response 8: We did not do a meta-analysis because the assumption of uniform population in studies across intervention and outcomes was violated. Some studies had used children with behavioural difficulties while other studies used children without any behavioural issues. We then decided to do a qualitative synthesis of the studies’ findings by presenting trends in figures 2 and 3 in our manuscript.
Round 2
Reviewer 2 Report
The text after authors revision is better.